# Electrospun Sesbania Gum-Based Polymeric *N*-Halamines for Antibacterial Applications

**DOI:** 10.3390/polym11071117

**Published:** 2019-07-01

**Authors:** Shi Lan, Yaning Lu, Jinghua Zhang, Yanan Guo, Chun Li, Shuang Zhao, Xianliang Sheng, Alideertu Dong

**Affiliations:** 1College of Science, Inner Mongolia Agricultural University, Hohhot 010018, China; 2College of Chemistry and Chemical Engineering, Inner Mongolia University, Hohhot 010021, China

**Keywords:** electrospinning, sesbania gum, polymeric *N*-halamine, naturally occurring polymer, antibacterial

## Abstract

Microorganism pollution induced by pathogens has become a serious concern in recent years. In response, research on antibacterial *N*-halamines has made impressive progress in developing ways to combat this pollution. While synthetic polymer-based *N*-halamines have been widely developed and in some cases even commercialized, *N*-halamines based on naturally occurring polymers remain underexplored. In this contribution, we report for the first time on a strategy for developing sesbania gum (SG)-based polymeric *N*-halamines by a four-step approach Using SG as the initial polymer, we obtained SG-based polymeric *N*-halamines (abbreviated as cSG-PAN nanofibers) via a step-by-step controllable synthesis process. With the assistance of advanced techniques, the as-synthesized cSG-PAN nanofibers were systematically characterized in terms of their chemical composition and morphology. In a series of antibacterial and cytotoxicity evaluations, the as-obtained cSG-PAN nanofibers displayed good antibacterial activity against *Escherichia coli* and *Staphylococcus aureus*, as well as low cytotoxicity towards A549 cells. We believe this study offers a guide for developing naturally occurring polymer-based antibacterial *N*-halamines that have great potential for antibacterial applications.

## 1. Introduction

Pathogenic bacteria thrive all around us—in drinking water, the air, our food, the soil, and so forth [1]. It has been demonstrated that many pathogenic bacteria cause significant microbial contamination that results in considerable economic burdens for society [2]. The specter of pandemics caused by pathogenic bacteria has raised global fears and prompted urgent investigations into preventing what could be some of the greatest challenges to public health in the near future [3,4]. Water disinfection, air purification, safe food packaging and storage, household sanitation, and public healthcare are all areas of major research in the quest to avert devastating mortality levels by containing or entirely eliminating pathogenic bacteria before they become hazardous [5,6,7,8,9,10].

Alexander Fleming’s discovery of penicillin in 1928 ushered in the antibiotic era [11]. However, the misuse of antibiotics has fueled antibiotic resistance, thereby limiting treatment options and making the rapid spread of more dangerous pathogenic bacteria an increasingly greater threat to human health [12,13]. Finding and developing sources that have novel antibacterial activity against pathogenic bacteria have drawn increasing research interest [14]. To date, a variety of antibacterial agents have evidenced their bactericidal and bacteriostatic qualities, including salt of hypochlorite, silver nanoparticles, zinc oxide, quaternary ammonium salts, peptides, povidone-iodines, *N*-halamines, and other antibacterial polymers [15,16,17,18,19,20,21,22]. *N*-halamines are powerful disinfectants against pathogenic bacteria-associated diseases [23]. Their numerous advantages over other antibacterial agents, including rapid action, high efficiency, long-term stability and durability, low price, and renewability, make them attractive in the areas of water decontamination, air purification, food storage and packaging, public healthcare, and textile safety, among others [24,25,26,27]. Accordingly, the past several years have witnessed the development of a number of antibacterial *N*-halamines, some of which have become commercially available [28].

Typically, *N*-halamines that include N–X (X = Cl, Br, or I) covalent bond(s) can easily be achieved via a *N*-halogenation reaction between their N–H bond-containing precursor and a halogenating agent (e.g., NaClO), bonding the oxidative halogen(s) onto the nitrogen [28]. *N*-halamines can thereby stabilize oxidative halogen(s) in N–X covalent bonds, providing much higher stability than traditional hypochlorite disinfectants [29]. *N*-halamines are more diverse in their molecular structures than other antibacterial agents [30], and they can be divided into two sub-categories according to their N–H bond-bearing precursors: low-molecular *N*-halamines and polymeric *N*-halamines [31]. The former, which include chloro derivatives of ammonia, CH_3_NHCl, and chlorosuccinimide, have a longer history of development, reaching back to 1927 [32]. In the early 1990s, Worley’s group began studying antibacterial polymeric *N*-halamines, finding they had excellent antibacterial properties [33]. Since then, a variety of polymeric *N*-halamines have been developed through bench research in academic laboratories, leading to practical applications and commercialization [34,35,36]. However, most scientists and industry researchers have focused their investigations on synthetic polymeric N-halamines; so far, chitosan-based *N*-halamines are the only type of naturally occurring polymeric *N*-halamines to receive attention [37]. Hence, finding and developing additional naturally occurring polymeric *N*-halamines is an important area for research.

Sesbania gum (SG) is a natural polysaccharide (a polymer) extracted from sesbania seeds [38]. As shown in Figure 1A, an SG molecule consists of two main parts: (i) glycosidic bonds linking mannose to form the main structure, and (ii) α (1→6) glycosidic bonds linking galactose on the side chains [39]. Because SG is natural, hydrophilic, low-cost, eco-friendly, and readily available, many researchers have concentrated on its versatility as a covalently functionalized support in a range of applications, such as catalysis, flocculation, pharmaceuticals, cosmetics, dyes and pigments, textiles, and adsorption [40,41]. Exploring the antibacterial potential of SG is also beginning to garner interest. Pal et al. grafted a cationic polymer onto SG via microwave-assisted graft copolymerization to achieve a material with antibacterial properties [42]. Our group introduced silver nanoparticles into a SG system using electrospinning to render SG with superior antibacterial activity [43]. As far as we know, however, no one has reported on experimental and theoretical research into the synthesis of SG-based antibacterial *N*-halamines.

Herein, we report for the first time on synthesizing electrospun SG-based polymeric *N*-halamines through a three-step reaction that combines epoxidation, amination, and chlorination, followed by electrospinning (Figure 1B). As illustrated in Figure 1C, SG was modified using a step-by-step process, and three products were achieved: epoxidized SG (eSG), aminated SG (aSG), and chlorinated SG (cSG). With plenty of N–Cl covalent bonds in its polymeric structure, the as-synthesized cSG had excellent antibacterial properties. To achieve greater antibacterial activity via the advantages of nanoscale adjustments, the cSG was then employed to prepare electrospun nanofibers (cSG-PAN; see Figure 1D). This strategy depends on the unprecedented synergism of SG modification, antibacterial chlorination, and electrospinning, yielding electrospun SG-based polymeric *N*-halamines with great potential for practical applications in antibacterial-related fields, water disinfection, air purification, textiles, medical devices and healthcare products, food storage and processing, and other applications.

## 2. Materials and Methods

### 2.1. Materials

Sesbania gum (SG) was obtained commercially from the Hongtu Plant Adhesive Factory (Qingdao, China). Polyacrylonitrile (PAN) was obtained from the Tianjin Chemical Reagent Plant (Tianjin, China). Sodium hydroxide (NaOH) and hydrochloric acid (HCl) were obtained from the Tianjin Chemical Reagent Plant (Tianjin, China). *N*,*N*-dimethylformamide (DMF), epichlorohydrin, triethylenetetramine, tetramethyl ammonium bromide, and sodium hypochlorite (NaClO) were purchased from Sinopharm Chemical Reagent Co., Ltd. (Shanghai, China). Deionized water was utilized for all of the experiments. All the chemicals were used without purification.

### 2.2. Synthesis of Epoxidized SG

About 1 g of SG was added into 36 mL of an aqueous solution of sodium hydroxide (NaOH), and the reaction was processed under stirring at room temperature. After stirring for 3 h, 1 mL of epichlorohydrin and 1 mL of an aqueous solution of tetramethyl ammonium bromide (1%) were added to the solution, then the reaction was continued under stirring for another 3 h. The pH value of the resulting solution was adjusted to 6–8, and the as-obtained epoxidized SG (eSG) was taken from the mixture solution, precipitated repeatedly with absolute alcohol, washed thoroughly with distilled water, and dried under vacuum.

### 2.3. Synthesis of Aminated SG

About 1.5 mL of triethylenetetramine was added slowly to the eSG solution under stirring, then the reaction was conducted at 60 °C for 12 h. The as-synthesized products were taken out and precipitated repeatedly with absolute alcohol to remove unreacted triethylenetetramine, then dried under vacuum.

### 2.4. Synthesis of Chlorinated SG

About 1 g of the as-prepared aSG was immersed in 50 mL of sodium hypochlorite (NaClO) solution, and the chlorination reaction was carried out for 4 h. Excess sodium hypochlorite was removed from the suspension by extensive dialysis in water, and the cSG was then dried at 60 °C for 1 h.

### 2.5. Electrospinning

The SG-based polymeric *N*-halamine nanofibers were prepared using a facile electrospinning method, which has been previously reported [44]. Typically, after 0.22 g of cSG was dispersed in 10 mL of DMF, about 1.18 g of PAN was then added, and the mixture was stirred overnight to obtain an achromatous, transparent, and viscous electrospinning precursor solution. Subsequently, the electrospinning procedure was run on an electrospinning unit at a high voltage of 12 kV and room temperature to obtain SG-based polymeric *N*-halamine nanofibers.

### 2.6. Characterization

^1^H-NMR spectra were recorded on a Bruker AVANCE III-500 instrument (Bruker, Co. Ltd., Billeria, MA, USA) in D_2_O solution. Fourier transform infrared (FTIR) spectrometry was recorded on a Thermo Nicolet Avatar 370 FTIR spectrometer (Thermo Fisher Scientific Co. Ltd., Waltham, MA, USA). X-ray photoelectron spectroscopy (XPS) was carried out on a PHI-5000CESCA system with Mg K radiation. Scanning electron microscopy (SEM) images (Shimadzu, Co. Ltd., Kyoto, Japan) were taken on a Shimadzu SSX-550 field emission scanning electron microscope at 15.0 kV. A drop of the well-dispersed sample in ethanol was cast onto a piece of silicon wafer and air dried. A thin gold coating was utilized to avoid charging during scanning, and a thorough microscopic study was carried out. Energy-dispersive X-ray (EDX) spectroscopy was also performed during the SEM measurements. The detailed morphology and size of the samples were observed with a Hitachi H-8100 transmission electron microscope (TEM, Hitachi, Ltd., Tokyo, Japan). The samples were well dispersed in ethanol with the assistance of sonication at 35 KHz, and the as-prepared dispersion was added dropwise onto the copper grid and dried at room temperature for TEM characterization.

### 2.7. Antibacterial Test

Antibacterial testing was performed using the colony-counting method [45], with *Escherichia coli* (*E. coli*, 8099, a typical Gram-negative bacterium) and *Staphylococcus aureus* (*S. aureus*, ATCC 6538, a typical Gram-positive bacterium) as the two model bacteria. Typically, *E. coli* and *S. aureus* were grown overnight at 37 °C in Luria–Bertani (LB) medium, then the bacterial cells were harvested by centrifugation, washed with phosphate-buffered saline, and diluted to concentrations of 1 × 10^6^ CFU/mL. Next, 50 μL of the bacterial suspension was mixed with 0.45 mL of the sample suspension (1 mg/mL) and incubated under constant shaking. After a certain period of contact time, the mixture was serially diluted, and 100 μL of each dilution was dispersed onto the LB growth medium. Survival colonies on the LB plates were counted after incubation for 24 at 37 °C. The colony-counting tests were carried out in triplicate. Bacterial survival was calculated according to the following equation:Bacterial survival = (*B*/*A*) × 100%
where *A* is the number of colonies in the control and *B* is the number of bacterial colonies surviving after treatment with the sample. 

### 2.8. Cytotoxicity Test

Briefly, A549 cells (8000 cells/well) were seeded into 96-well plates and incubated in 10% FBS-containing DMEM overnight under a standard cultivation environment of 5% CO_2_ at 37 °C. For the cytotoxicity test, various concentrations of samples (10, 30, 60, 150, 250 and 500 μg/mL) were added into the wells. After incubating for 24 h, 20 μL of MTT solution (5 mg/mL) was added into each well that was incubated at 37 °C for 4 h. Finally, dimethyl sulfoxide (DMSO, 120 μL per well) was used to dissolve the formed formanzan crystal, and the absorbance at 492 nm was measured by GF-M3000 microplate reader (CAIHONG, Shandong, China). The cell proliferation (%) was calculated as the ratio of sample to control, which represented the absorbance values of the treated and untreated cells, respectively.

## 3. Results and Discussion

The electrospun SG-based polymeric *N*-halamines were fabricated via a combined epoxidation–amination–chlorination–electrospinning approach. Figure 1C,D displays the synthesis procedure. First, SG reacted with epichlorohydrin in the presence of tetramethyl ammonium bromide to achieve epoxidized SG (eSG). Subsequently, the surface of the eSG was chemically modified with triethylenetetramine, forming aminated SG (aSG). Then, chlorinated SG (cSG) was obtained by a chlorination reaction between aSG and sodium hypochlorite, coupled with N–H → N–Cl transformation. Finally, using polyacrylonitrile (PAN) as a support matrix, the cSG was turned into electrospun nanofibers, yielding the final products of electrospun SG-based polymeric *N*-halamines (cSG-PAN). The incorporation of *N*-halamines (i.e., cSG) into the PAN-supported electrospun system conferred antibacterial capability upon the final nanofibers. This combined epoxidation–amination–chlorination–electrospinning strategy is also effective for developing other functional materials.

The formation of SG-based polymeric *N*-halamines was detected using FTIR spectroscopy. Figure 2 shows the FTIR spectra of SG, eSG, aSG, and cSG. For pristine SG, the characteristic peaks at 3454, 2932, and 2840 cm^−1^ are attributed to the stretching vibrations of the –OH and C–H bands, respectively [41]. Two peaks at 1647 and 1440 cm^−1^ are assigned to the C=O stretching vibration and –OH bending vibration, respectively [46]. Two characteristic peaks at 1157 and 1076 cm^−1^ are due to the C–O stretching vibration [47]. Compared to pristine SG, eSG displayed epoxy absorption band at 879 cm^−1^ and weaker –OH stretching vibrations at 3454 cm^−1^, indicating the success of the epoxidation treatment on the –OH groups of SG [41]. In the case of aSG, the two obvious absorption bands at 2928 and 2849 cm^–1^ are assigned to the stretching vibrations of the C–H band. Obviously, these two peaks are stronger than those of eSG, demonstrating that there were more C–H bands on aSG than on eSG and thereby confirming the reaction between triethylenetetramine and eSG. After chlorination was applied to aSG, the as-synthesized cSG retained the characteristic peaks corresponding to –OH and C–H bands. There were no significant differences between the cSG and aSG spectra, which suggests that, except for the N–H → N–Cl transformation, the chlorination reaction between aSG and sodium hypochlorite had no impact on the chemical structure of aSG.

The morphologies and compositions of the as-produced samples were examined using a SEM technique combined with the EDX spectra. Figure 3 presents SEM images and the corresponding EDX spectra of (A) SG, (B) eSG, and (C) cSG. It is clear that the three products possess different morphologies and structures. Within the selected SEM region, SG (A1) displayed many small, irregular particles with smooth surfaces. In contrast, eSG (B1) showed an aggregated morphology and a wrinkled, loose surface, suggesting a morphological evolution from pristine SG to eSG. After the amination–chlorination treatment, the as-formed cSG (C1) presented an aggregate morphology with quite a rough surface. Detailed comparison indicates obvious differences between the morphologies and surface states of SG, eSG, and cSG, demonstrating the morphological changes resulting from the step-by-step synthesis of cSG. To acquire chemical compositional information, the EDX spectra of SG, eSG, and cSG were examined at selected sites (red “+” letters) on the SEM images. Pristine SG (A2) displays two intensive peaks assigned to C and O signals, suggesting that the SG consisted mainly of carbon and oxygen. In the case of eSG (B2), an intensive Al peak is clear, arising from the aluminum foil used for sample immobilization. The Na peak corresponds to sodium hydroxide, used as the alkaline medium during the epoxidation reaction of epichlorohydrin with SG. Notably, the eSG showed a higher O/C ratio than pristine SG, indicating the effectiveness of the epoxidation treatment. Unlike pristine SG and eSG, the cSG presented two typical peaks corresponding to N and Cl, indicating the N–Cl bond on the surface of SG and the success of both the amination and the chlorination reactions. As can be seen from Figure 3C1, the cSG sample was coated thickly on the surface of the aluminum foil, so the corresponding signal of aluminum foil is not strong enough to be detected.

To further confirm the effectiveness of the four-step combined synthesis method, detailed information about the chemical composition of the as-synthesized products was gathered using XPS spectra. Figure 4 presents XPS survey scans, showing the C 1s spectrum, O 1s spectrum, N 1s spectrum, and Cl 2p spectrum of (A) SG, (B) aSG, (C) cSG, and (D) cSG-PAN. The two elemental peaks at 154 and 103 eV are attributed, respectively, to Si 2s and Si 2p, both of which can be seen in all four samples. The appearance of these two peaks marks the presence of SiO_2_, which is attributed to the glass support applied for sample immobilization [48]. The pristine SG (A) exhibits two main peaks, those of C 1s and O 1s, indicating that SG is made up of elemental carbon and oxygen. After the epoxidation–amination treatment, the as-obtained aSG (B) shows an additional peak for N 1s, which is effective proof of the formation of aSG [40]. With cSG (C), the four typical signals of C 1s, O 1s, N 1s, and Cl 2p are clearly visible [49], suggesting the N–H bonds were transformed into N–Cl bonds when chlorination was carried out with aSG. When aSG was mixed with PAN in DMF, followed by electrospinning, the as-electrospun cSG-PAN also shows the four characteristic peaks of C 1s, O 1s, N 1s, and Cl 2p, confirming that the electrospinning has almost no impact on the chemical composition of the aSG. Comparison between aSG and aSG-PAN shows that the latter has the more intensive N 1s signal, demonstrating the successful combination of PAN with aSG.

Electrospinning is an effective and facile process for synthesizing fibers with sizes in the range of micrometers to nanometers [50]. The morphology and size of the electrospun products depends on many parameters, such as the concentration of electrospinning precursor solution, distance between needle and target collector, voltage, feed rate of the solution, solvent, etc. [51]. Among these, the solvent is one of the most decisive parameters. To date, many previous reports have shown that the morphology and size of the electrospun products are solvent-dependent [52]. Tungprapa et al. investigated the effects of the solvent system on the morphological appearance and size of cellulose acetate (CA) products [53]. Tiyek et al. studied the impact of the solvent system on the fiber surface morphology and hydrophilicity of electrospun polycaprolactone (PCL) [54]. In most cases, the solubility and spinnability of sample in solvent(s) are two main factors influencing the morphology and size of the electrospun products. Taking solubility and spinnability into consideration, N, N-dimethylformamide (DMF) was utilized herein to fabricate electrospun cSG-PAN.

Morphological information about the electrospun cSG-PAN formed via electrospinning was obtained using SEM. Figure 5 presents SEM images of the cSG-PAN prepared with different feed ratios of cSG to PAN. Unlike cSG, which showed an irregular morphology and a rough surface (Figure 3), the cSG-PAN after electrospinning showed a fiber-like morphology with a smooth surface, suggesting the feasibility and effectiveness of electrospinning to prepare SG-based polymeric *N*-halamine nanofibers. For comparison, pure PAN was also spun, resulting in a randomly oriented, straight, and continuous, fiber-like regular morphology. Although most of the cSG-PAN had a straight, fiber-like morphology, some bulges and spindles were randomly scattered amongst the fibers. Magnified SEM images are provided to clarify the surface state of the cSG-PAN nanofibers, showing their glossy appearance. To further verify whether the cSG particles were distributed evenly in the PAN polymer matrix, TEM was employed to observe the inner structure of the cSG-PAN nanofibers. Figure 6 presents TEM images of cSG-PAN nanofibers prepared with different feed ratios of cSG to PAN. The nanofibers were evenly distributed without any aggregation. Even in the spindles (indicated by the red arrows in Figure 6), there was no aggregation, demonstrating that the appearance of the cSG-PAN was due to the mixture of cSG and PAN rather than any aggregation or uneven distribution of cSG in the PAN matrix.

After confirming the success of our electrospinning-guided synthesis, we then regulated the synthesis of cSG-PAN by tuning the feed ratio of cSG to PAN. In our step-by-step synthesis, the feed ratio of cSG to PAN was changed from 1:5 to 1:10, then to 1:20, and the impacts of the different feed ratios on the morphology and size of cSG-PAN were determined (Figure 5 and Figure 6). Overall, no morphological changes were detected; all three samples presented fiber-like morphological features, suggesting the feed ratio had no influence upon the morphology of cSG-PAN.

As shown in Figure 7, the average sizes of the cSG-PAN nanofibers prepared with different feed ratios of cSG to PAN were determined by measuring the sizes of 30 randomly selected cSG-PAN nanofibers. When the feed ratio of cSG to PAN was decreased from 1:5 to 1:10, then to 1:20, the average diameters for three cSG-PAN nanofibers were 173, 276, and 552 nm, respectively, demonstrating that the size of cSG-PAN nanofibers increased with the feed ratio of cSG to PAN. It can be concluded that the size of electrospun SG-based polymeric *N*-halamines can be regulated simply by tuning the feed ratio of cSG to PAN. When we further examined the size distribution of the cSG-PAN nanofibers, it was clear that all three nanofibers had narrow size distributions—70–270 nm for a 1:5 feed ratio, 140–410 nm for a 1:10 feed ratio, and 280–820 nm for a 1:20 feed ratio—further indicating the validity of the controlled synthesis strategy for electrospun SG-based polymeric *N*-halamines. In addition to the feed ratio of cSG to PAN, the effect of the vertical distance between the buret tip and target collector on the morphology and dimension was investigated. When other parameters were kept constant, the vertical distance increased from 12.5, to 14.5, to 16.5, to 18.5, then to 20.5 cm, respectively. As shown in Figure 8, all five cSG-PAN products show the fiber-spindle morphology, suggesting that vertical distance has no impact on nanofiber morphology. However, the nanofiber diameter decreases with the increase of vertical distance, demonstrating that the lengthening vertical distance could result in the decrease of the diameter of the cSG-PAN nanofibers. The increase of journey of cSG-PAN jet might be an important explanation for the decrease of nanofiber diameter.

Afterwards, we examined the antibacterial activities of the cSG-PAN nanofibers (with the active chlorine of 0.37 Cl^+^ %) against a common pathogenic bacterium, *E. coli*. In our first set, an inhibition zone study was employed to confirm that the cSG-PAN nanofibers had bactericidal capability. Figure 9A–C presents photographs of the inhibition zone test. There were clear aseptic halos around the three cSG-PAN samples, showing that it inhibited *E. coli*. Since it is well known that PAN has no antibacterial qualities [55], the antibacterial activity of the cSG-PAN nanofibers can be attributed to cSG rather than PAN. The appearance of inhibition rings around the cSG-PAN nanofibers not only confirms that the N–Cl bonds endowed the electrospun nanofibers with bactericidal capacity but also proves it is possible to modify SG polymers with *N*-halamines to make them effective antibacterial agents.

The relationship between the active chlorine content of the cSG-PAN and its antibacterial activity was studied. Figure 9D shows the inhibition zone of the cSG-PAN nanofibers with the active chlorine content of 0.91 Cl^+^ %. Obviously, 0.91 Cl^+^ % product presents a larger inhibition zone than 0.37 Cl^+^ % product (Figure 9A), indicating that the higher the active chlorine content is, the more active cSG-PAN nanofibers will be. To confirm the long-term stability and antibacterial durability, the cSG-PAN nanofibers were stored at 25 °C and 65% for one month and washed with water, and then their antibacterial activities were estimated using the inhibition zone test. As shown in Figure 9E,F, the inhibition zones are close to those before storage and the washing test, suggesting that the cSG-PAN nanofibers have high stability and antibacterial durability.

In addition to confirming antibacterial capability, the inhibition zone test is also an effective tool to confirm release-based antibacterial action [56]. The appearance of inhibition rings around the cSG-PAN nanofibers indicates that some active chlorines (i.e., Cl^+^) were released from the cSG-PAN nanofibers by the dissociation of the N–Cl bonds and the released chlorines attacked the bacteria [57]. To confirm this release action, we applied the iodometric/thiosulfate test [58]. Two oxidation–reduction reactions below are involved in this test.
(1)$$\begin{array}{l} N-Cl+2I^{-}+H^{+}\to N-H+I_{2}+Cl^{-}\\ I_{2}+2S_{2}O_{3}^{2-}\to 2I^{-}+S_{4}O_{6}^{2-} \end{array}$$

Prior to testing, the cSG-PAN powder was suspended in water for about 24 h and centrifuged at 10,000 rpm, then the supernatant was tested using iodometric/thiosulfate reactions. We found that the supernatant could oxidize iodide ions to produce iodine, confirming the release of active chlorine from cSG-PAN in the water system. When a detailed comparison was done, three cSG-PAN nanofibers showed different degrees of inhibition activity; the diameters of the inhibition rings were 19.2 mm for the 1:5 feed ratio, 18.1 mm for the 1:10 feed ratio, and 15.9 mm for the 1:20 feed ratio. The cSG loading on the cSG-PAN nanofibers was the probable reason for the variations in inhibition level.

To better understand the antibacterial action of the cSG-PAN nanofibers, the colony-counting method was used with *E. coli* and *S. aureus* as two model bacteria. The antibacterial effects were gauged after culturing the treated bacteria on LB agar subsequent to 60 min of exposure to the cSG-PAN nanofibers. As shown in Figure 10A, the surviving bacterial colonies on the culture plates were detectable as small white dots [59]. The control group cultured in the absence of the cSG-PAN nanofibers displayed robust growth. In contrast, serious decreases were seen in the *E. coli* and *S. aureus* colonies in the presence of the cSG-PAN nanofibers, indicating the fibers had bactericidal abilities against the two model bacteria.

The susceptibility of the bacteria to the cSG-PAN nanofibers was investigated through a time–kill assay, by which the rate and extent of antibacterial action could be determined. In our assays, we studied the time-dependent antibacterial actions of cSG-PAN nanofibers against *E. coli* and *S. aureus*. Figure 10B shows the time–kill curves of three cSG-PAN nanofiber samples against *E. coli* and *S. aureus*. Initially, bacterial survival decreases rapidly, then it slows down as the contact period lengthens, which agrees well with the trends of other antibacterial agents reported previously [59]. Moreover, the nanofibers’ antibacterial impact gradually changed from bacteriostatic to bactericidal with the change in aging time. Accordingly, we conclude that the antibacterial action of the cSG-PAN nanofibers, as well as their biocidal efficiency against bacteria, can be regulated simply by tuning the contact time between nanofibers and bacteria.

As for clinical use, antibacterial materials always need good biocompatibility and no toxicity to normal cells [60]. Accordingly, the cytotoxicity of the cSG-PAN nanofibers was examined herein through the MTT assay using A549 cells as model cells. SG and PAN were used as two comparatives, and their cytotoxicities were tested as well. For the MTT cell viability assay (Figure 11), all three samples (SG, PAN, and cSG-PAN nanofibers) exhibit high cell survivals within the whole sample concentration range from 10 μg/mL to 500 μg/mL, demonstrating that they possess negligible cytotoxicity towards A549 cells. Compared to pristine SG and PAN, the cSG-PAN nanofibers display lower cell survivals, suggesting that the cSG-PAN nanofibers are much more cytotoxic to A549 cells than both SG and PAN. However, the A549 cells after treated with 500 μg/mL of the cSG-PAN nanofibers show >80% survival, which indicates that the cSG-PAN nanofibers are safe enough and potent for biomedical use.

## 4. Conclusions

In summary, novel SG-based polymeric *N*-halamine electrospun nanofibers were synthesized via a combined epoxidation–amination–chlorination–electrospinning approach. By using a series of advanced techniques, including FTIR spectra, XPS spectra, SEM and TEM microscopy, and EDX spectra, the as-synthesized cSG-PAN nanofibers were systematically characterized in terms of their compositions and morphologies, and their synthesis was regulated by tuning the feed ratio of cSG to PAN. We found that the feed ratio of cSG to PAN played a significant role in controlling the nanofibers’ chemical composition and antibacterial activity. After active chlorine was loaded onto the polymer nanofibers, the final cSG-PAN nanofibers showed excellent antibacterial activity against *E. coli* and *S. aureus*. We proposed the possible mechanism of the as-obtained cSG-PAN nanofibers’ antibacterial action. These systematic investigations of SG-based polymeric *N*-halamine electrospun nanofibers point to new directions for exploring the use of naturally occurring polymer-based antibacterial *N*-halamines in antibacterial applications.

## Figures and Tables

**Figure 1 polymers-11-01117-f001:**
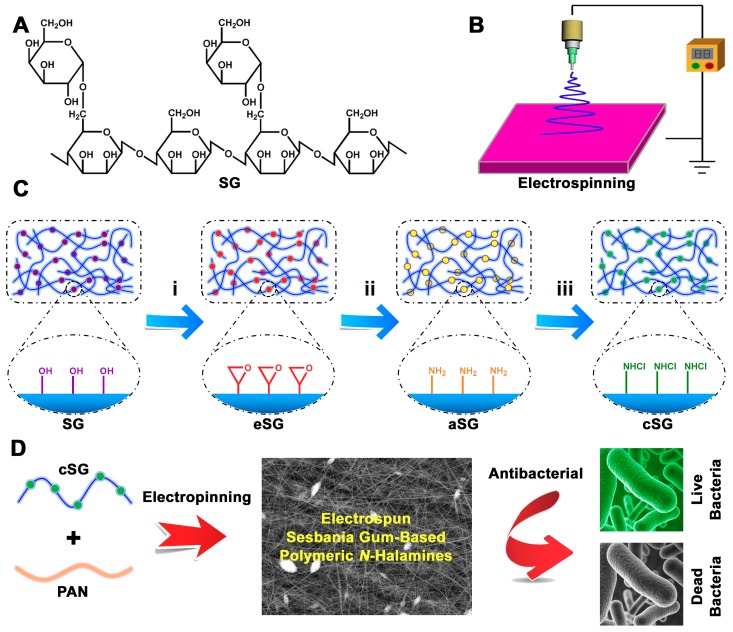
(**A**) Chemical structure of sesbania gum (SG). (**B**) Schematic diagram of the electrospinning device. (**C**) Schematic illustration of sesbania gum-based polymeric *N*-halamine. (**D**) Schematic depiction of the synthesis of SG-based polymeric *N*-halamines (cSG-PAN) using electrospinning and the antibacterial action of cSG-PAN.

**Figure 2 polymers-11-01117-f002:**
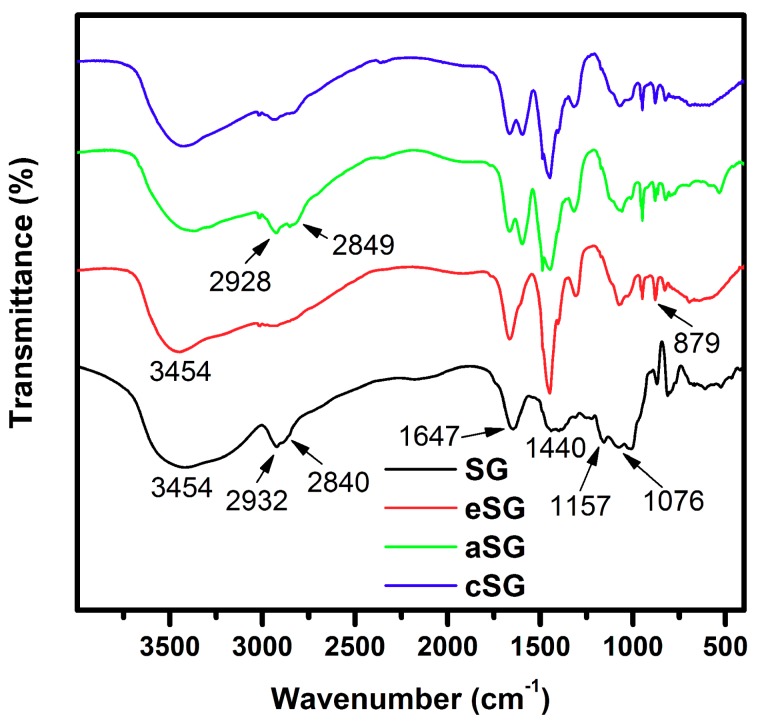
FTIR spectra of SG, eSG, aSG, and cSG.

**Figure 3 polymers-11-01117-f003:**
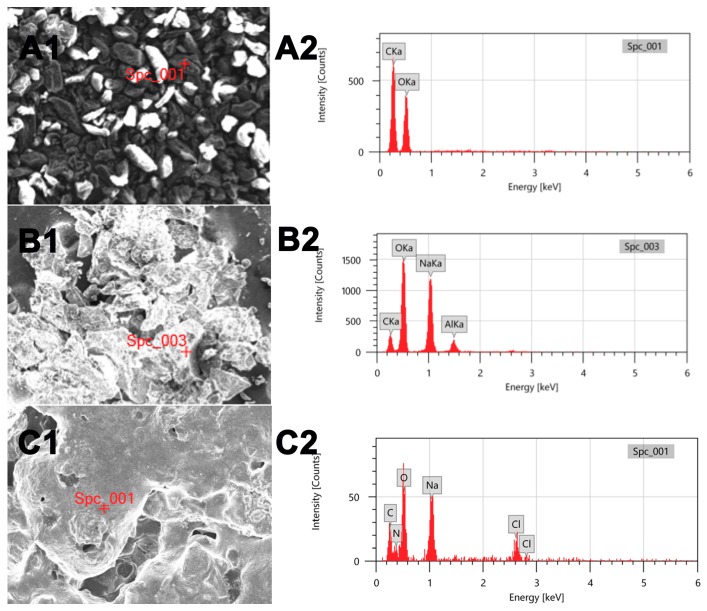
SEM images and EDX spectra of (**A1,A2**) SG, (**B1,B2**) eSG, and (**C1,C2**) cSG.

**Figure 4 polymers-11-01117-f004:**
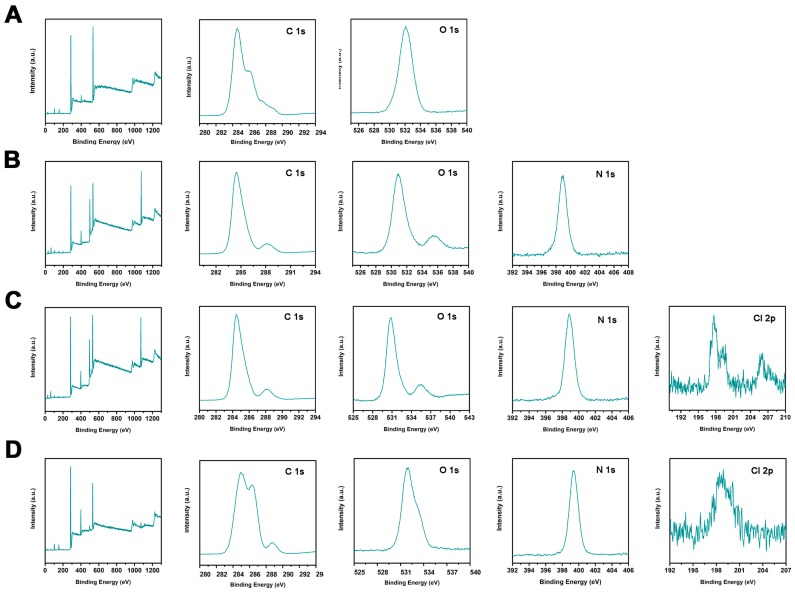
XPS survey scans: C 1s spectrum, O 1s spectrum, N 1s spectrum, and Cl 2p spectrum of (**A**) SG, (**B**) aSG, (**C**) cSG, and (**D**) cSG-PAN.

**Figure 5 polymers-11-01117-f005:**
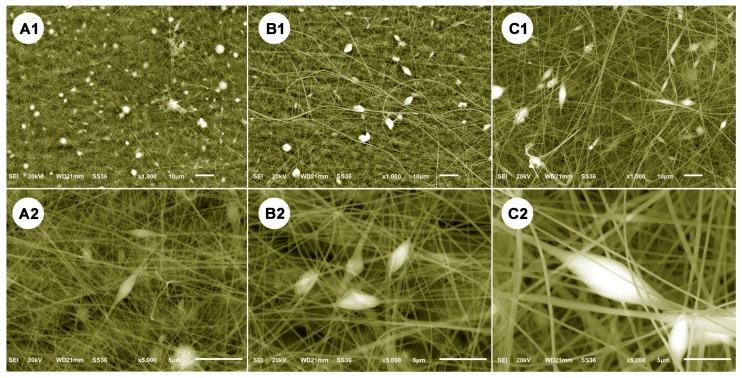
SEM images of cSG-PAN prepared with different feed ratios of cSG to PAN: (**A1**,**A2**) 1:5, (**B1**,**B2**) 1:10, and (**C1**,**C2**) 1:20.

**Figure 6 polymers-11-01117-f006:**
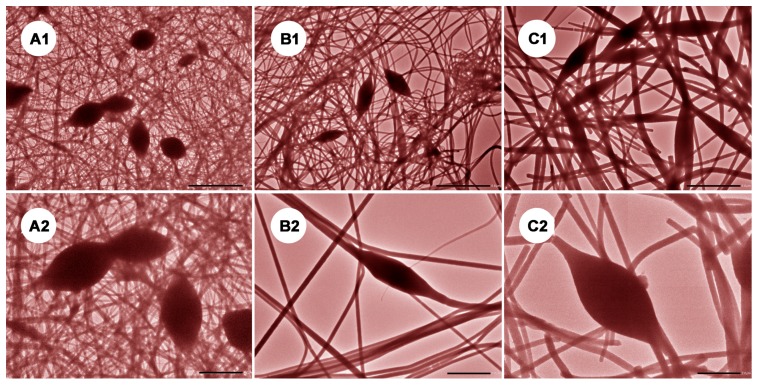
TEM images of cSG-PAN prepared with different feed ratios of cSG to PAN: (**A1**,**A2**) 1:5, (**B1**,**B2**) 1:10, and (**C1**,**C2**) 1:20.

**Figure 7 polymers-11-01117-f007:**
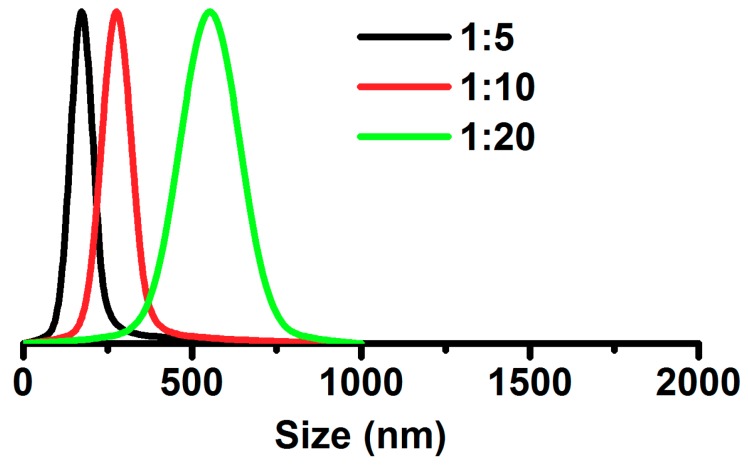
Size distributions of cSG-PAN prepared with different feed ratios of cSG to PAN.

**Figure 8 polymers-11-01117-f008:**
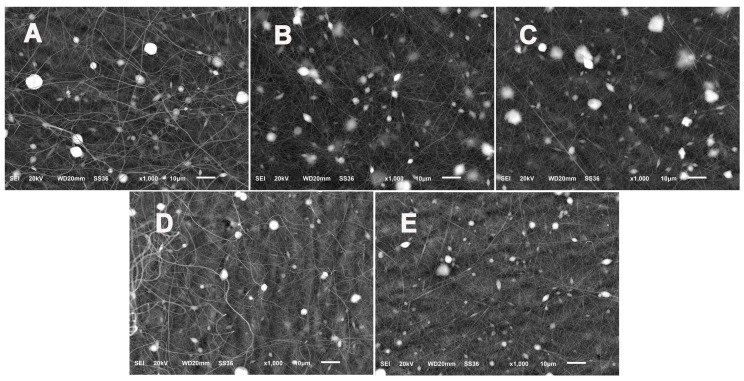
SEM images of cSG-PAN prepared with different vertical distance between the buret tip and collector: (**A**) 12.5; (**B**) 14.5; (**C**) 16.5; (**D**) 18.5; (**E**) 20.5 cm.

**Figure 9 polymers-11-01117-f009:**
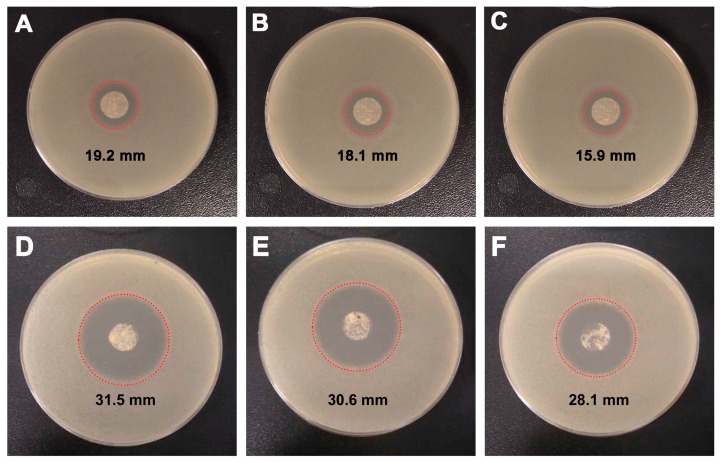
(**A**–**C**) Photographs of the inhibition zone of cSG-PAN prepared with different feed ratios of cSG to PAN against *E. coli*: (**A**) 1:5; (**B**) 1:10; and (**C**) 1:20. (**D**–**F**) Photographs of the inhibition zone of cSG-PAN (**D**) before and (**E**) after one month of storage at 25 °C and 65% RH and (**F**) after washing durability test.

**Figure 10 polymers-11-01117-f010:**
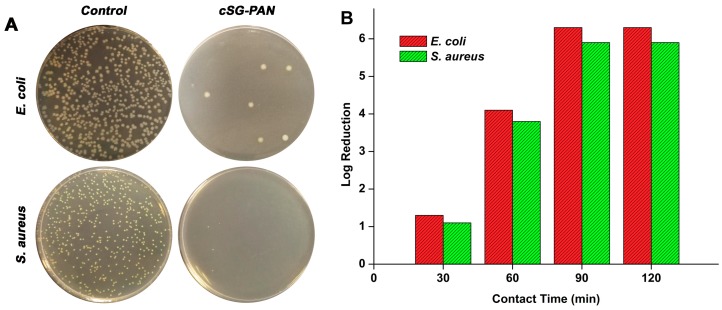
(**A**) Photographs of culture plates of *E. coli* and *S. aureus* in the absence and presence of cSG-PAN nanofibers. (**B**) Time–kill assay with cSG-PAN nanofibers against *E. coli* and *S. aureus*.

**Figure 11 polymers-11-01117-f011:**
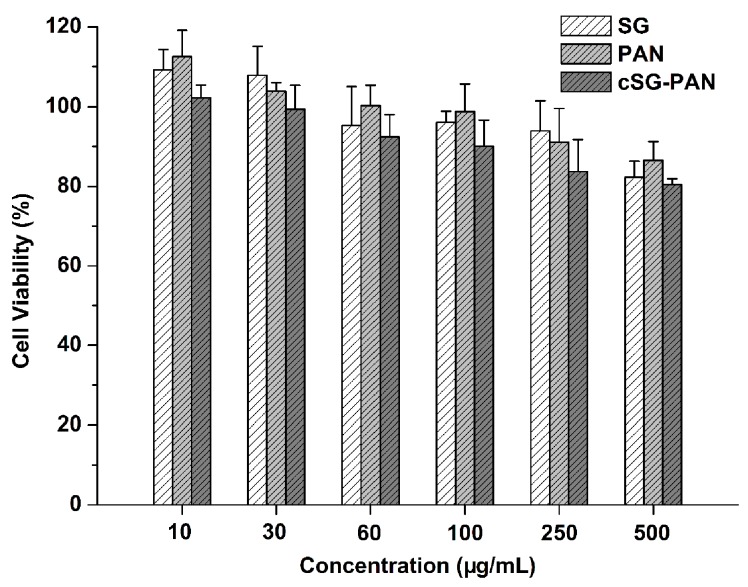
Survival percentage of A549 cells after treated with SG, PAN, and cSG-PAN nanofibers.

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
