# Peer review of "Electrospun Sesbania Gum-Based Polymeric N-Halamines for Antibacterial Applications"

_polymers, 2019, doi:10.3390/polym11071117_

Round 1
Reviewer 1 Report
The theme of the article is very interesting and important, taking into account the current problem in relation to the resistance of bactéria. It is urgente to find new strategies to com bat the microorganisms. This work is about a strategy for developing naturally occurring polymer-based antibacterial N-halamines. I recommend the publication of this article in "Polymers" after major revision. The subjects that the authors must improve are:
- what are the applications for this type of samples?
- There is no analysis and discussion in FTIR spectra in the range of 1000-2000 cm-1
- In fig. 3 C2 I do not see the peak of NKa
- The authors should explain how the samples were prepared for TEM
- The authors should analyze the toxicity of the samples and the duration of bacterial activity
Author Response
Reviewer #1:
The theme of the article is very interesting and important, taking into account the current problem in relation to the resistance of bacteria. It is urgent to find new strategies to combat the microorganisms. This work is about a strategy for developing naturally occurring polymer-based antibacterial N-halamines. I recommend the publication of this article in “Polymers” after major revision. The subjects that the authors must improve are:
Comment 1: What are the applications for this type of samples?
Response : This study shows the unprecedented synergism of SG modification, antibacterial chlorination, and electrospinning, yielding electrospun SG-based polymeric N-halamines with great potential for practical applications in antibacterial-related fields, including: water disinfection, air purification, textiles, medical devices and healthcare products, food storage and processing, and other applications.
The as-synthesized SG-based polymeric N-halamines can be used in a form of antibacterial additives. Typically, the SG-based polymeric N-halamines can be introduced into functional materials, e.g., polymer matrixs, to endow them antibacterial function. In contrast to bulk counterparts, the SG-based polymeric N-halamines can offer higher antibacterial activity due to their smaller size and higher surface area. Also, the SG-based polymeric N-halamines can also be utilized as antibacterial coatings on materials’ surfaces by the aid of chemical bindings. For example, the SG-based polymeric N-halamines can be coated on the surface of medical devices, textiles, and plastics, which makes their surfaces effective in fighting against pathogenic bacteria, as well as in preventing disease infection. Additionally, the SG-based polymeric N-halamines can also be employed in water disinfection and air purification. By adding into polymeric membranes (e.g., cotton, cellulose, etc.), the SG-based polymeric N-halamines could render membranes antibacterial properties. The combination of permeability and antibacterial allow them usable in filtration system, such as water disinfection and air purification.
According to the reviewer suggestion, the potential applications for the as-synthesized SG-based polymeric N-halamines have been provided in the revised manuscript.
Comment 2: There is no analysis and discussion in FTIR spectra in the range of 1000-2000 cm-1.
Response : Based on the reviewer suggestion, the FTIR spectra have been discussed systematically in the revised manuscript.
Comment 3: In fig. 3 C2 I do not see the peak of N Ka.
Response : As reviewer pointed out, the N Ka signal is not clear in Fig. 3C2. So, the EDX spectrum of cSG was re-measured, and a new representative image has been given in the revised manuscript.
Comment 4: The authors should explain how the samples were prepared for TEM
Response : According to the reviewer suggestion, more detailed information about the preparation of samples for TEM measurements have been provided in the revised manuscript.
Comment 5: The authors should analyze the toxicity of the samples and the duration of bacterial activity.
Response : According to the reviewer suggestion, the toxicity of the as-prepared cSG-PAN nanofibers and the duration of bacterial activity was analyzed in the revised manuscript.

Reviewer 2 Report
The authors present a study whereby a polysaccharide was electrospun into fibers and surface functionalized with biocidal N-halamine groups. The fibers kill bacteria by releasing oxidative species into solution, which rapidly inactive Gram-positive and Gram-negative bacteria.
I recommend publication after the authors address the following:
Are these materials toxic to human cells? They show killing of bacteria, but it's also important to assess the selectivity of action based on cell type.
In Figure 9, sterilization rate (%) is shown as a function of time It would be more informative to show CFU/mL on a log scale versus time, so readers can observe the log reduction. It is not so useful to have a linear scale of % because 99.9% and 99.99% look the same on such a plot - whereas 3-log versus 4-log reduction is a big difference.
The introduction points to a review article on antibacterial polymers from 2012. Many great and more current reviews have been published since then, including: Biomacromolecules, 2018, 196, 1888-1917 and Polymer Chemistry 2018, 9, 2407-2427
Author Response
Reviewer #2:
The authors present a study whereby a polysaccharide was electrospun into fibers and surface functionalized with biocidal N-halamine groups. The fibers kill bacteria by releasing oxidative species into solution, which rapidly inactive Gram-positive and Gram-negative bacteria. I recommend publication after the authors address the following:
Comment 1: Are these materials toxic to human cells? They show killing of bacteria, but it’s also important to assess the selectivity of action based on cell type.
Response : According to the reviewer suggestion, the cytotoxicity of the as-prepared cSG-PAN nanofibers was examined and the corresponding descriptions have been provided in the revised manuscript.
Comment 2: In Figure 9, sterilization rate (%) is shown as a function of time. It would be more informative to show CFU/mL on a log scale versus time, so readers can observe the log reduction. It is not so useful to have a linear scale of % because 99.9 % and 99.99 % ook the same on such a plot - whereas 3-log versus 4-log reduction is a big difference.
Response : As reviewer said, the efficiencies of antibacterial agents are commonly evaluated by detecting the log reduction of bacteria in the presence of antibacterial agents. Therefore, the sterilization rate (%) in original Fig. 10B has been replaced by log reduction in the revised manuscript.
Comment 3: The introduction points to a review article on antibacterial polymers from 2012. Many great and more current reviews have been published since then, including: Biomacromolecules, 2018, 196, 1888-1917 and Polymer Chemistry 2018, 9, 2407-2427.
Response : According to the reviewer suggestion, several typical literatures (Biomacromolecules, 2018, 19, 1888; Polym. Chem., 2018, 9, 2407; J. Mater. Chem. B, 2018, 6, 7217; Macromol. Mater. Eng., 2014, 299, 648; ACS Infec. Dis., 2017, 3, 293; Int. J. Antimicrob. Agent., 2015, 46, 446; Adv. Colloid Interface Sci., 2014, 203, 37) have been cited in the revised manuscript.

Reviewer 3 Report
In this paper for the first time was reported a strategy for developing sesbania gum (SG)-based polymeric N-halamines by a four-step approach. Using SG as the initial polymer, SG-based polymeric N-halamines (abbreviated as cSG-PAN nanofibers) were obtained via a step-by-step controllable synthesis process. To achieve greater antibacterial activity via the advantages of nanoscale modifications, the cSG was then employed to prepare electrospun nanofibers. This strategy depends on the unique synergism of SG modification, antibacterial chlorination, and electrospinning, yielding electrospun SG-based polymeric N-halamines with great potential for practical applications bactericidal agent.
The paper is interesting and discussion is well-written; however the authors need to address the issue listed below before re-submission of this paper.
Major changes:
1. The authors should discuss how influence of the distance between needle and target has on the morphology and nanofiber dimension.
2. The charge of the deposited fibers has influence of the morphology of the fibers but there are other parameters such as solvent, which also contribute. The authors should make some comments related to the solvent factor on morphology of electrospun fibers. Some examples should be provided.
Minor changes:
1. Major peaks in FTIR need to be labelled.
Author Response
Reviewer #3:
In this paper for the first time was reported a strategy for developing sesbania gum (SG)-based polymeric N-halamines by a four-step approach. Using SG as the initial polymer, SG-based polymeric N-halamines (abbreviated as cSG-PAN nanofibers) were obtained via a step-by-step controllable synthesis process. To achieve greater antibacterial activity via the advantages of nanoscale modifications, the cSG was then employed to prepare electrospun nanofibers. This strategy depends on the unique synergism of SG modification, antibacterial chlorination, and electrospinning, yielding electrospun SG-based polymeric N-halamines with great potential for practical applications bactericidal agent. The paper is interesting and discussion is well-written; however the authors need to address the issue listed below before re-submission of this paper.
Comment 1: The authors should discuss how influence of the distance between needle and target has on the morphology and nanofiber dimension.
Response : According to the reviewer suggestion, the impact of vertical distance between buret tip and target collector on the morphology and dimension has been examined and discussed in the revised manuscript.
Comment 2: The charge of the deposited fibers has influence of the morphology of the fibers but there are other parameters such as solvent, which also contribute. The authors should make some comments related to the solvent factor on morphology of electrospun fibers. Some examples should be provided.
Response: As reviewer stated, the morphology of the electrospun products depends on many parameters, such as concentration of electrospinning precursor solution, distance between buret tip and target collector, voltage, feed rate of the solution, solvent, etc. Among them, solvent is one of the most decisive parameters. In fact, the solubility and spinnability of sample in solvents are two main parts influencing the product morphology.
According to the reviewer suggestion, some comments related to the solvent factor on morphology of electrospun products have been added in the revised manuscript. Also some typical literatures (Eur. Polym. J. 2005, 41, 409-421; J. Polym. Sci. B: Polym. Phys. 2004, 42, 5-11; Bull. Mater. Sci. 2019, 42, 171; Cellulose 2007, 14, 563-575; J. Nanosci. Nanotechnol. 2019, 19, 7251-7260) have been cited in the revised manuscript.
Comment 3: Major peaks in FTIR need to be labeled.
Response : According to the reviewer’s suggestion, the main peaks in FTIR spectra have been labeled in the revised manuscript.

Round 2
Reviewer 1 Report
I thank the authors have made the corrections suggested by me.
The authors respond well to my questions, I have only one question, in the figure 3 C2 EDX spectrum of cSG it has to be done again, because in the revised manuscript the EDX spectrum has very few counts and the peaks of Al and Si desappeared, what happened?
Author Response
Reviewer #1: I thank the authors have made the corrections suggested by me. Comment 1: The authors respond well to my questions, I have only one question, in the figure 3 C2 EDX spectrum of cSG it has to be done again, because in the revised manuscript the EDX spectrum has very few counts and the peaks of Al and Si disappeared, what happened? Response : Thanks for your kind suggestion. As reviewer said, the EDX spectrum of cSG has few counts and the peaks of Al and Si disappeared. The possible reason is as follows. The sample powders of cSG were grinded well and dispersed on aluminium foil for EDX measurement. As can be seen from SEM image of cSG (Figure 3C1), the cSG sample was coated thickly on the surface of the aluminium foil, so the corresponding signal of Al (and Si) element is not strong enough to be tested. As a result, we could find several main peaks including C, O, N, Na, and Cl element in EDX spectrum of cSG.

Reviewer 2 Report
The authors responded satisfactorily to all issues raised in the first round of reviews. I can now recommend publication in the present form.
Author Response
Reviewer #2: Comment : The authors responded satisfactorily to all issues raised in the first round of reviews. I can now recommend publication in the present form. Response : Thanks for your kind help and suggestions.

Reviewer 3 Report
The authors amended the manuscript as per reviwer's comments. The revised manuscript is acceptable for publication.
Author Response
Reviewer #3: Comment : The authors amended the manuscript as per reviwer’s comments. The revised manuscript is acceptable for publication. Response : Thanks for your kind help and suggestions.
